# Transcription Factor Lmx1b Negatively Regulates Osteoblast Differentiation and Bone Formation

**DOI:** 10.3390/ijms23095225

**Published:** 2022-05-07

**Authors:** Kabsun Kim, Jung Ha Kim, Inyoung Kim, Semun Seong, Jeong Eun Han, Keun-Bae Lee, Jeong-Tae Koh, Nacksung Kim

**Affiliations:** 1Department of Pharmacology, Chonnam National University Medical School, Gwangju 61469, Korea; kabsun@hanmail.net (K.K.); kjhpw@hanmail.net (J.H.K.); doll517@naver.com (I.K.); iamsemun@gmail.com (S.S.); 2Hard-Tissue Biointerface Research Center, School of Dentistry, Chonnam National University, Gwangju 61186, Korea; jtkoh@chonnam.ac.kr; 3Department of Orthopedic Surgery, Chonnam National University Medical School and Hospital, Gwangju 61469, Korea; oshanje@gmail.com (J.E.H.); kbleeos@chonnam.ac.kr (K.-B.L.); 4Department of Pharmacology and Dental Therapeutics, School of Dentistry, Chonnam National University, Gwangju 61186, Korea

**Keywords:** Lmx1b, BMP2, Runx2, osteoblast, bone formation

## Abstract

The LIM-homeodomain transcription factor Lmx1b plays a key role in body pattern formation during development. Although Lmx1b is essential for the normal development of multiple tissues, its regulatory mechanism in bone cells remains unclear. Here, we demonstrated that Lmx1b negatively regulates bone morphogenic protein 2 (BMP2)-induced osteoblast differentiation. Overexpressed Lmx1b in the osteoblast precursor cells inhibited alkaline phosphatase (ALP) activity and nodule formation, as well as the expression of osteoblast maker genes, including runt-related transcription factor 2 (Runx2), alkaline phosphatase (Alpl), bone sialoprotein (Ibsp), and osteocalcin (Bglap). Conversely, the knockdown of Lmx1b in the osteoblast precursors enhanced the osteoblast differentiation and function. Lmx1b physically interacted with and repressed the transcriptional activity of Runx2 by reducing the recruitment of Runx2 to the promoter region of its target genes. In vivo analysis of BMP2-induced ectopic bone formation revealed that the knockdown of Lmx1b promoted osteogenic differentiation and bone regeneration. Our data demonstrate that Lmx1b negatively regulates osteoblast differentiation and function through regulation of Runx2 and provides a molecular basis for therapeutic targets for bone diseases.

## 1. Introduction

Osteoblasts are responsible for maintaining the balance in bone metabolism via the bone formation in cooperation with osteoclasts, which are responsible for bone resorption. Functional disorders of osteoclasts and osteoblasts, especially the abnormal increase in the activity of osteoclasts, are responsible for many diseases, such as osteoporosis, osteolysis, and Paget’s disease [1,2,3]. Osteoblasts that differentiate from mesenchymal stem cells synthesize the bone matrix and organize the mineralization of the skeleton for bone formation.

Bone morphogenic protein 2 (BMP2) is an important signaling ligand for osteoblast differentiation, which functions by activating the Smad signaling pathway [4]. When activated, Smad1/5/8 forms a heteromeric complex with Smad4, which is translocated into the nucleus and interacts with several transcription factors such as runt-related transcription factor 2 (Runx2), osterix (Osx), distaless5 (Dlx5), and meshless2 [5,6]. Runx2 is a master transcription factor involved in osteogenic gene expression and osteoblast differentiation. Runx2 binds to the osteoblast-specific cis-acting element (OSE2) in the promoter region of various osteogenic genes, including alkaline phosphatase (Alpl), bone sialoprotein (Ibsp), and osteocalcin (Bglap), and collagen type I alpha1 (Col1α1) [7,8]. Upon activation of the BMP signaling cascade, Runx2 and Smads physically interact with each other to regulate the transcription of the target genes cooperatively and thereby enhance osteoblast differentiation [9].

The LIM homeodomain (HD) transcription factor 1 beta (Lmx1b) belongs to a subfamily of the LHX transcription factor genes. Lmx1b contains two N-terminal cysteine and histidine rich zinc-binding LIM domains which mediate its interaction with other proteins, a central homeodomain responsible for the interaction of Lmx1b with its target genes, and a C-terminal glutamine rich transcriptional activation domain [10,11]. Lmx1b is required for the regulation of dorsal-ventral patterning of the limbs and in organ development, including the kidney, brain, and eye [12,13], and mutations in Lmx1b were identified in patients with nail patella syndrome (NPS) [14]. In addition, Lmx1b knockout mice exhibit embryonic lethality, kidney abnormalities, calvarial bone defects, an absence of the cerebellum, and dorsal to ventral conversion of limbs [15,16]. Although many studies have focused on the function of Lmx1b in development and differentiation, the role of Lmx1b in bone remodeling remains unclear.

In this study, we aimed to elucidate the molecular pathways regulated by Lmx1b; we focused on the identification of partner genes that interact with Lmx1b in osteoblasts and the importance of Lmx1b in osteoblast differentiation and function. We identified the molecular cascade mediated by Lmx1b during BMP2-induced bone formation, which revealed that Lmx1b suppresses Runx2 activity via interaction with Runx2 during osteoblast differentiation. Furthermore, knockdown of Lmx1b promoted osteogenic differentiation in vitro and BMP2-induced ectopic bone formation in vivo. We demonstrated that Lmx1b negatively regulates the transcription of Runx2 target genes in cooperation with Runx2 during osteoblast differentiation and new bone formation.

## 2. Results

### 2.1. Lmx1b Negatively Regulates Osteoblast Differentiation

We investigated whether Lmx1b is expressed in osteoblasts and osteoclasts. First, to determine whether Lmx1b is expressed during osteoblast differentiation, primary osteoblast precursor cells obtained from calvaria of neonatal mice were cultured in the presence of osteogenic media (OGM), including ascorbic acid, β-glycerophosphate, and BMP2. Real-time PCR analyses showed that the expression of osteoblast marker genes such as *Runx2*, *Alpl*, *Ibsp*, and *Bglap* was significantly elevated during the osteoblast differentiation, and *Lmx1b* was continuously expressed in osteoblast differentiation (Figure 1a). Next, to confirm that Lmx1b is expressed in osteoclasts, osteoclast differentiation was induced by culturing bone marrow-derived monocyte/macrophage lineage cells in the presence of macrophage colony stimulating factor (M-CSF) and receptor activator of nuclear factors κB ligand (RANKL). In the RT-PCR analysis, unexpectedly, Lmx1b expression was not detected during osteoclast differentiation (Appendix A).

To investigate the role of Lmx1b in osteoblasts, primary preosteoblasts were infected with the control or Lmx1b retrovirus and further cultured with OGM. Compared to the overexpression of the control vector, overexpression of Lmx1b in osteoblasts strongly inhibited ALP activity and mineralized nodule formation under OGM (Figure 1b,c). Overexpression of Lmx1b significantly reduced the expression of the osteogenic genes *Runx2*, *Alpl*, *Ibsp*, and *Bglap* during osteoblast differentiation as compared to the overexpression of the control vector, at both the mRNA (Figure 1d) and protein levels (Figure 1e). Taken together, these results indicate that Lmx1b negatively regulates osteoblast differentiation.

### 2.2. Knockdown of Lmx1b Promotes Osteoblast Differentiation

To further confirm the role of Lmx1b, *Lmx1b* silencing using siRNA was performed to downregulate *Lmx1b* expression during osteoblast differentiation. Primary osteoblast precursor cells transfected with *Lmx1b*-specific siRNA were cultured with OGM. *Lmx1b*-specific siRNA-transfected preosteoblasts showed significant enhancement in ALP activity and mineralized nodule formation compared to the control siRNA cells (Figure 2a,b). As expected, knockdown of *Lmx1b* significantly increased the expression of *Runx2*, *Alpl*, *Ibsp*, and *Bglap* (Figure 2c). Taken together, these data support the negative role of Lmx1b in osteoblast differentiation and function.

### 2.3. Regulation of Runx2 Transcriptional Activity by Lmx1b

Since we observed that Lmx1b regulates Runx2 expression on osteoblast differentiation through loss-of function and gain-of-function experiments, we investigated the effect of Lmx1b on the transcriptional regulation of Runx2. Runx2 is known as a master regulator of bone development, and Runx2 binds osteoblast specific cis acting element (OSE2) in promoters of various osteogenic genes. We performed the luciferase reporter assay to assess the role of Lmx1b in the transcriptional activity of Runx2 using Runx2-responsive reporters. When a reporter plasmid was transfected with Runx2 and Lmx1b, Lmx1b significantly suppressed the Runx2-dependent transcriptional activation of Ocn and Bsp reporter, respectively (Figure 3a,b). To determine whether Lmx1b affects the binding of Runx2 to the Runx2-responsive promoter region for downregulation of Runx2 transcriptional activity, a chromatin IP (ChIP) assay was performed using the cultured mature osteoblasts. As shown in Figure 3c, Lmx1b suppressed the recruitment of Runx2 to the promoters of Runx2 target genes such as *Alpl*, *Ibsp*, and *Bglap*. Furthermore, recruitment of Runx2 to the Runx2 target genes increased following the inhibition of Lmx1b expression (Figure 3d). To investigate whether Lmx1b could interact with Runx2, we transfected 293T cells with HA-Runx2 or Flag-Lmx1b and performed a co-immunoprecipitation assay. As shown in Figure 3e,f, Lmx1b showed physical interactions with Runx2. Consistent with this result, endogenous Lmx1b and Runx2 showed interactions in differentiated osteoblasts (Figure 3g). Collectively, these results suggest that Lmx1b inhibits osteoblast differentiation by interacting with Runx2 and repressing Runx2 transcriptional activity.

### 2.4. Lmx1b Regulates BMP2-Induced Signaling Pathways in Osteoblasts

We next examined the role of Lmx1b in BMP2-induced signaling pathways such as Smads and mitogen-activated protein kinases (MAPKs). Primary osteoblast precursors were infected with the control or Lmx1b retrovirus and followed by BMP2 stimulation. As shown in Figure 4a, treatment with BMP2 increased Smad1/5/8, AKT, and ERK phosphorylation in osteoblasts infected with the control vector; however, AKT phosphorylation was strongly inhibited, while Smad1/5/8 and ERK were attenuated in Lmx1b overexpressed osteoblasts. Conversely, the activation of Smad1/5/8, AKT, and ERK was promoted after stimulation by BMP2 in osteoblasts showing Lmx1b downregulation compared to that in the control cells (Figure 4b). Taken together, these results indicate that Lmx1b also regulates BMP2-mediated signaling pathways during osteoblast differentiation.

### 2.5. Lmx1b Knockdown Can Enhance BMP2-Induced Ectopic Bone Formation In Vivo

It has been established that BMP2 plays an important role in bone tissue engineering, and can induce ectopic bone formation [17]. We investigated the role of Lmx1b in BMP2-induced ectopic bone formation with subcutaneous implantation of a BMP2-containing collagen sponge, which is a well-established animal model for BMP2-induced ectopic bone formation [18,19]. Collagen sponges containing control siRNA with BMP2, or *Lmx1b* siRNA with BMP2 were delivered into the subcutaneous space into the back of the mice on both sides. After three weeks of implantation, new bone formation was analyzed using high resolution micro computed tomography (µCT) analysis and histology. µCT analysis revealed that implantation of the collagen sponge containing *Lmx1b* siRNA with BMP2 significantly promoted the extent of bone formation compared to that shown by control siRNA with BMP2, and histological analysis showed that the area of new generated bones was higher in mice with a collagen sponge containing *Lmx1b* siRNA with BMP2 (Figure 5a). As shown in Figure 5b, bone parameters such as bone volume fraction (BV/TV) and bone volume significantly increased in the collagen sponge containing *Lmx1b* siRNA with the BMP2 group compared to that in the control siRNA with BMP2 group. We observed a significant reduction in Lmx1b expression levels and increased expression levels of *Ibsp* and *Bglap* in ectopic bone specimens treated with *Lmx1b* siRNA with BMP2 compared to that in the group treated with control siRNA with BMP2 (Figure 5c). These findings indicate that Lmx1b regulates osteoblast differentiation in vitro and bone formation in vivo. Taken together, our data suggest that Lmx1b plays an important role in bone generation.

## 3. Discussion

Lmx1b is essential for the normal development of dorsal limb structures, the glomerular basement membrane, the anterior segment of the eye, and dopaminergic and serotonergic neurons [12,14,15,20]. Mutations of Lmx1b in humans are generally associated with an autosomal dominant inherited disease called the nail-patella syndrome, which is characterized by developmental defects in the dorsal limb structures, such as the absence of small patellae, joint abnormalities with elbow contractures, and the iliac horns of the pelvis [21]. Although Lmx1b knockout mice reportedly exhibit calvarial bone defects, the role of Lmx1b in osteoblasts has not been fully elucidated yet. In this study, we showed that Lmx1b decreases osteoblast differentiation and function by regulating Runx2 activity.

Runx2 is a transcription factor that is essential during early stage osteoblast differentiation and osteoblast commitment and cooperates with numerous transcription factors and cofactors [22]. The transcriptional activity of Runx2 is dependent on its nuclear localization, where it binds to the OSE2 on the promoters of osteogenic genes including Alpl, Bglap and Ibsp. Runx2 activity is precisely regulated by interactions with transcriptional factors. For example, BMP-responsive Smads, TAZ, Ets, Dlx5, Hes1, and KLF2 enhance the transcriptional activity of Runx2 target genes through physical interaction with Runx2 [23,24,25,26,27,28]. Meanwhile, Rev-erbα, Stat1, Nrf2, MEF, and TWIST are associated with Runx2 and inhibit Runx2 transactivation by sequestering Runx2 in the cytoplasm and interfering with the binding of Runx2 to Runx2-responsive promoter [29,30,31,32,33]. In the present study, we demonstrated that Lmx1b was capable of physical interaction with Runx2, which subsequently blocked the recruitment of Runx2 to Runx2-responsive promoter during osteoblast differentiation.

In addition, BMP2, a major inducer of osteoblast differentiation is involved in a cascade of downstream signal transduction, such as Smad1/5/8, AKT, and ERK signaling, upon binding to its receptor [4,34]. Our results demonstrated that Lmx1b regulated the phosphorylation of BMP2-mediated signaling pathways. However, ERK activation was not effectively induced by BMP2 compared to that of other signaling pathways. Therefore, the magnitude of inhibition or elevation of ERK activation by Lmx1b overexpression or Lmx1b silencing, respectively, was not immense as those of other signaling pathways. Furthermore, it is well known that the p38 MAPK signal is present downstream of the BMP2–BMPR signaling pathway. However, the phosphorylation of p38 MAPK was not altered by either Lmx1b overexpression or Lmx1b silencing; therefore, it is deduced that Lmx1b does not regulate the entire BMP2 signaling pathway during osteoblast differentiation (Appendix A). In addition, we investigated whether the expression of BMP receptors was affected by Lmx1b. As depicted in Appendix A, the expression levels of *Bmpr1α*, *Bmpr1β*, and *BmprII* were not altered by either Lmx1b overexpression or Lmx1b knockdown, suggesting that Lmx1b did not affect the expression of BMP receptors in the regulation of signaling associated with osteoblast differentiation. Although Lmx1b suppressed the BMP2 signaling pathway in part, the specific mechanism underlying this phenomenon is still elusive and requires further investigation. Collectively, the present study suggests that Lmx1b negatively regulates osteoblast differentiation via physical interaction with Runx2 and is involved in the regulation of BMP2-mediated signaling pathways.

Through loss and gain of function experiments, previous reports have shown that Lmx1b plays an important role in specifying the potential of different regions within the head mesenchyme with different osteogenic at the beginning of calvarial bone development but not during limb development [16,35]. Cesario et al. showed that ectopic expression of Lmx1b in the supraorbital mesenchyme inhibits bone formation; inactivation of Lmx1b in the head mesenchyme induced expression of osteogenic marker genes such as Runx2, Sp7, and Dlx5 in the vertex mesenchyme and expanded the bone-forming area, resulting in synostosis [16]. Although the role of Lmx1b was restricted to the area of bone formation within the calvarial primordium, these reports are consistent with our results that Lmx1b suppresses primary calvarial osteoblast differentiation and function; therefore, these reports support our finding that Lmx1b might act as an anti-osteoblastogenic gene.

Recent studies have identified via yeast two-hybrid assays that CLIM2 (LDB1) and PAX2 interact directly with LMX1B; in addition, immunoprecipitation (IP) experiments followed by mass spectrometry analyses have identified PSPC1, GRLF1, DDX9, MYO1C, HSP70, and TMPO (LAP2α) as possible interactors for binding to LMX1B. Among them, it had been reported that Pax2 promoted osteogenic differentiation by regulating Runx2 transcriptional activity via MAPK pathways [36]. In addition, studies have shown that HSP70 enhances the osteogenic differentiation of human mesenchymal stem cell through mediation of the transcription factors Runx2 and Osx via activation of the ERK signaling pathway [37]. A recent study reported that TMPO (LAP2α) promoted the osteogenic differentiation of human adipose derived stem cells by modulating NF-kB signaling [38]. Therefore, further investigation is necessary to explore whether Lmx1b is involved in regulating the functions of its well-known binding interactors such as Pax2, Hsp70, and TMPO during osteogenic differentiation.

In addition, we demonstrated that the knockdown of Lmx1b promoted osteogenic differentiation and bone regeneration in BMP2-induced ectopic bone formation model in vivo. Taken together, our data suggest that Lmx1b acts as a negative regulator of Runx2 in BMP2 signaling pathway in osteoblasts. Further studies on the regulation of Lmx1b are needed to determine its therapeutic potential in bone regeneration.

## 4. Materials and Methods

### 4.1. Reagents

Recombinant human BMP2 was purchased from Cowellmedi (Busan, Korea). Alizarin Red, β-glycerophosphate, and p-nitrophenyl phosphate were purchased from Sigma (St. Louis, MO, USA). Ascorbic acid was purchased from Junsei Chemical (Tokyo, Japan).

### 4.2. Constructs

Lmx1b was cloned using RT-PCR with RNA obtained from mature osteoblasts. The primer sequences are as follows; Lmx1b 5′-ATG TTG GAC GGC ATC AAG A-3′ and 5′-CAG AGC TCC TAC TTT GCC TCC-3′. The amplified Lmx1b was subcloned either into pMX-IRES-EGFP or pcDNA3.1.

### 4.3. Retroviral Infection

Plat-E cells were maintained in Dulbecco’s modified Eagle’s medium (DMEM; HyClone Laboratories, Lagan, UT, USA) supplemented with 10% fetal bovine serum (FBS), 100 U/mL penicillin, and 100 µg/mL streptomycin. To prepare the retroviral supernatants, recombinant plasmids and the parental pMX vector were transfected into the packaging cell line Plat-E using Fugene6 (Promega, Madison, WI, USA) according to the manufacturer’s protocol. Viral supernatant was collected from cultured medium 48 h after transfection. Osteoblast precursor cells were incubated with the viral supernatants for 6 h in the presence of 10 µg/mL polybrene (Sigma).

### 4.4. Osteoblast Differentiation

Primary osteoblast precursor cells were isolated from the calvarias of newborn one-day-old mice by enzymatic digestion with α-MEM (Hyclone) containing 0.1% collagenase (Life Technologies, Carlsbad, CA, USA) and 0.2% dispase II (Roche Diagnostics GmbH, Mannheim, Germany). After the enzymes were removed, the collected cells were cultured in α-MEM (Hyclone) containing 10% FBS, 100 U/mL penicillin, and 100 μg/mL streptomycin. Osteoblast differentiation was promoted by incubation in osteogenic medium containing 100 ng/mL BMP2, 50 μg/mL ascorbic acid, and 100 μM β-glycerophosphate for 2 to 6 days, and the culture medium was replaced every 2 days (Lee et al., 2015; Kim et al., 2021). For the ALP activity assay, osteoblast precursor cells were lysed with an osteoblast lysis buffer (50 mM NaCl, pH 7.6, 150 mM NaCl, and 0.1% Triton X-100 and 1 mM EDTA). The cell lysates were incubated with p-nitrophenyl phosphate substrate (Sigma), and ALP activity was measured with a spectrophotometer at 405 nm. For Alizarin red staining, cells were cultured for six days, fixed with 70% ethanol, and stained with 40 mM Alizarin Red (pH 4.2) at room temperature. After removal of the nonspecific staining using PBS, the Alizarin Red staining was visualized using a CanoScan 9000F (Canon Inc., Tokyo, Japan). Following this, for quantification, stained Alizarin Red was dissolved in 10% cetylpyridium (Sigma) for 15 min at room temperature, and Alizarin Red’s activity was measured with a spectrophotometer at 562 nm.

### 4.5. Quantitative Real-Time PCR Analysis

Cells were lysed in Qiazol (Qiagen GmbH, Hilden, Germany), and total RNA was isolated from the cultured cells according to the manufacturer’s protocol. Purified RNA was reverse transcribed into cDNA using a QuantiNova Reverse Transcription Kit (Qiagen), including the procedure for removal of contaminating genomic DNA according to the manufacturer’s instructions. To examine the mRNA expression levels, the cDNA synthesized was used for SYBR-based real-time PCR performed in triplicate using a Rotor-Gene6 instrument (Qiagen). The thermal cycling conditions were as follows: 15 min at 95 °C, followed by 40 cycles at 94 °C for 15 s, 55 °C for 30 s, and 72 °C for 30 s. The mRNA concentrations were normalized to an endogenous housekeeping gene, Gapdh. The relative quantitation value for each target gene compared with the calibrator for the target was expressed as 2-(Ct-Cc) (Ct and Cc are the mean threshold cycle differences after normalizing to Gapdh). The relative expression levels of samples were presented as a semi-log plot. The primer sequences were as follows: Lmx1b 5′-CTG TCG CAA GGG TGA CTA-3′ and 5′-AGC AGC GAA GAG TTT CAA G-3′; Runx2 5′-CCC AGC CAC CTT TAC CTA CA-3′ and 5′-CAG CGT CAA CAC CAT CAT TC-3′; Alpl 5′-CAA GGA TAT CGA CGT GAT CAT G-3′ and 5′-GTC AGT CAG GTT GTT CCG ATT C -3′; Ibsp 5′-GGA AGA GGA GAC TTC AAA CGA A-3′ and 5′-CAT CCA CTT CTG CTT CTT CGT TC-3′; Bglap 5′-ATG AGG ACC CTC TCT CTG CTC AC-3′ and 5′-CCA TAC TGG TTT GAT AGC TCG TC-3′; Gapdh, 5′-TGA CCA CAG TCC ATG CCA TCA CTG-3′ and 5′-CAG GAG ACA ACC TGG TCC TCA GT G-3′.

### 4.6. Small Interfering RNA Transfection

Control siRNA and Lmx1b siRNA were purchased from Dharmacon (Lafayette, CO, USA) and transfected into preosteoblasts using Lipofectamine RNAiMAX (Thermo Fisher Scientific, Waltham, MA, USA) according to the manufacturer’s protocol. The knockdown level was verified using quantitative PCR.

### 4.7. Western Blot Analysis and Immunoprecipitation

After the osteoblast precursor cells were cultured for the indicated times, the cells were washed with chilled PBS and lysed in extraction buffer (50 mM Tris-HCl [pH 8.0], 150 mM NaCl, 1 mM EDTA, 0.5% Nonidet P-40, PMSF, and protease inhibitors). For immunoprecipitation, 293T cells were transfected with Flag-Lmx1b and HA-Runx2 for 48 h, or mature osteoblasts were harvested after washing with ice-cold PBS and lysed in extraction buffer. The lysates were immunoprecipitated using the indicated antibodies. The cell lysates and precipitated samples were subjected to sodium dodecyl sulfate polyacrylamide gel electrophoresis (SDS-PAGE) transferred to poly vinylidene fluoride (PVDF) membranes (Millipore, Billerica, MA, USA). The primary antibodies used included Runx2, p-Smad158 and Smad158 (Santa Cruz Biotechnology, Dallas, TX, USA); HA; Flag and Actin (Sigma); p-Akt, Akt, p-Erk, and Erk (Cell Signaling Technology, Danvers, MA, USA); and Alp, Osx, and Lmx1b (Abcam, Cambridge, UK). HRP-conjugated secondary antibodies (Abcam) were probed and developed with ECL solution (Millipore). Signals were detected using enhanced chemiluminescence and analyzed using the Azure c300 chemiluminescent western blot imaging system (Azure Biosystems Inc., Dublin, CA, USA).

### 4.8. Luciferase Assay

293T cells were cultured in DMEM (Hyclone) supplemented with 10% FBS, 100 U/mL penicillin, and 100 μg/mL streptomycin. Cells were transfected with the indicated amounts of expression plasmids using Fugene6 (Promega) following the manufacturer’s protocol. After 48 h of transfection, the cells were lysed in reporter lysis buffer (Promega) and analyzed using a luciferase assay system (Promega) according to the manufacturer’s instructions. Luciferase activity was measured in triplicate, averaged, and then normalized to β-galactosidase activity using ο-nitrophenyl-β-D-galactopyranoside (Sigma) as a substrate.

### 4.9. Chromatin Immunoprecipitation (ChIP) Assay

A ChIP assay was performed using the ezChIP kit (Millipore) according to the manufacturer’s instructions, using antibodies against Runx2, or control IgG (Santa Cruz Biotechnology). The precipitated DNA was subjected to PCR amplification with primers specific for the Alp or Bsp promoter region containing Runx2-binding sites. The following primers were used for PCR: Alp-p: 5′-GGC TGG GAC AGA CAG AAT GT-3′ and 5′-GTC CCT CGA TGG TTG T-3′; Bsp-p 5′-GCC TCA GTT GAA TAA ACA TGA AA-3′ and 5′-TTG TGG GAT TTA ATT GAA GGG TGA GGA-3′; Ocn-p: 5′-GAC AGC AAC AAT GTA TTC ATG -3′ and 5′-GCT CCT CAC ACT CTG AAA CC-3′. Real-time PCR was used to quantify ChIP assay results. All test Ct values were normalized using the input Ct value, and data were represented as fold enrichment.

### 4.10. Ectopic Bone Formation

Male mice (four-weeks-old) obtained from the Institute of Cancer Research (ICR) were randomly assigned to each experimental group. The mice were anesthetized by intraperitoneal injection with 0.1% Avertin (Sigma). For the ectopic bone formation model, BMP2-treated collagen sponges with control siRNA or *Lmx1b* siRNA mixed with Lipofectamine RNAiMAX (Thermo Fisher Scientific) were implanted under the dorsal skins of anesthetized mice (*n* = 7 per group) [17]. After four weeks, all mice were euthanized with CO_2_ to evaluate bone generation.

### 4.11. Osteoclast Differentiation

Murine osteoclasts were prepared from bone marrow cells, which were obtained by flushing the femurs and tibiae from six-week-old male ICR mice. The cells were cultured in α-MEM (Hyclone) containing and 30 ng/mL M-CSF for three days, and the attached cells were used as an osteoclast precursor. To generate osteoclasts, osteoclast precursor cells were cultured with M-CSF (30 ng/mL) and RANKL (100 ng/mL) for two days.

### 4.12. Statistical Analysis

Statistical analyses were performed using an unpaired Student’s *t*-test. All data are presented as the mean ± SD. *p* < 0.05 was considered statistically significant.

## Figures and Tables

**Figure 1 ijms-23-05225-f001:**
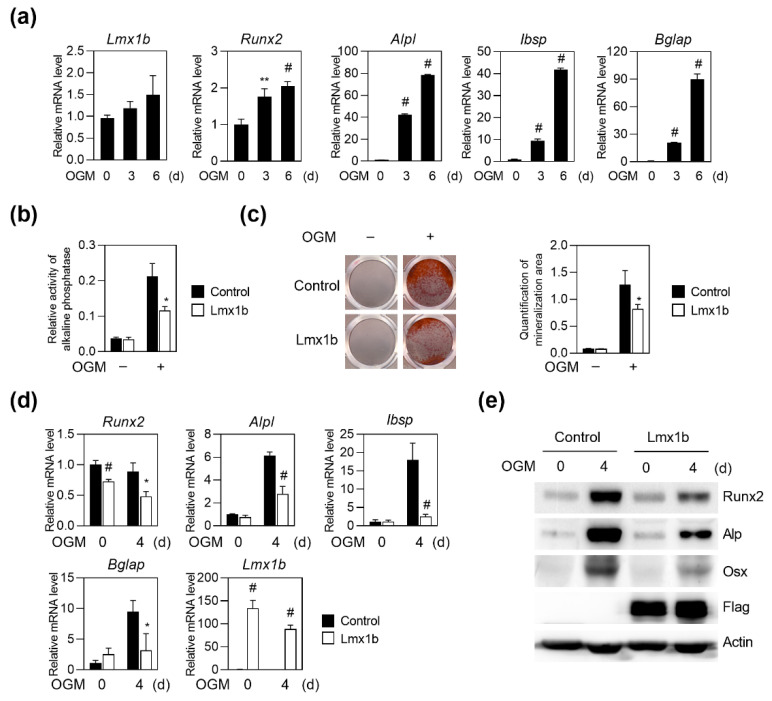
Lmx1b overexpression inhibits osteoblast differentiation and function. Primary osteoblast precursor cells were cultured in OGM (osteogenic medium) for six days. (**a**) Total RNA was isolated from cell lysates at the indicated times, and the mRNA expression levels of *Lmx1b*, *Runx2*, alkaline phosphatase (*Alpl*), bone sialoprotein (*Ibsp*), and osteocalcin (*Bglap*) were determined using quantitative real-time PCR. Data represent the means ± SD of triplicate samples. ** *p* < 0.001, # *p* < 0.005 vs. day 0. Values shown are normalized to *Gapdh* levels. Data represent means ± SD of triplicate samples. ** *p* < 0.001, # *p* < 0.005 vs. 0 day. Results are representative of experiments that were independently repeated at least three times. (**b**–**e**) Primary osteoblast precursor cells transduced with pMX-IRES-EGFP (Control) or Lmx1b retrovirus were cultured in OGM. (**b**) Cells cultured for four days were analyzed for alkaline phosphatase (ALP) activity. Data represent the means ± SD of triplicate samples. * *p* < 0.05 vs. control. (**c**) Cells cultured for six days were fixed and stained with alizarin red. Alizarin red staining activity quantified by densitometry at 570 nm. Data represent the means ± SD of triplicate samples. * *p* < 0.05 vs. Control. (**d**) Total RNA was isolated, and the mRNA expression levels of *Lmx1b*, *Runx2*, *Alpl*, *Ibsp*, and *Bglap* were determined using quantitative real-time PCR. Data represent the means ± SD of triplicate samples. * *p* < 0.05, # *p* < 0.005 vs. control. Results are representative of experiments that were independently repeated at least three times. (**e**) Cell lysates obtained at each time point were analyzed using western blotting with specific antibodies, as indicated.

**Figure 2 ijms-23-05225-f002:**
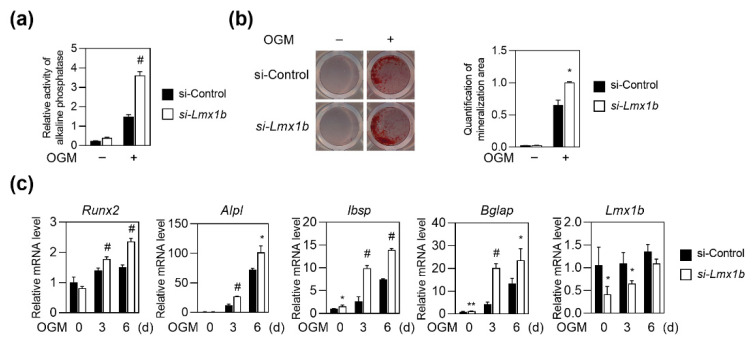
Knockdown of Lmx1b enhances osteoblast formation. (**a**–**c**) Primary osteoblast precursor cells transfected with control siRNA (si-Control) or siRNA specific for Lmx1b (*si-Lmx1b*) were cultured in OGM. (**a**) Cells cultured for four days were analyzed for ALP activity. Data represent the means ± SD of triplicate samples. # *p* < 0.05 vs. si-Control. Results are representative of experiments that were independently repeated at least three times. (**b**) Cells cultured for six days were stained with Alizarin Red. Alizarin Red’s staining activity was quantified by densitometry at 570 nm. Data represent the mean ± SD of triplicate samples. * *p* < 0.05 vs. si-Control. Results are representative of experiments that were independently repeated at least three times. (**c**) Total RNA was isolated and mRNA expression levels of *Runx2*, *Alpl*, *Ibsp*, *Bglap*, and *Lmx1b* were determined using quantitative real-time PCR. Data represent the mean ± SD of triplicate samples. * *p* < 0.05, ** *p* < 0.001, # *p* < 0.005, vs. si-Control. Results are representative of experiments that were independently repeated at least three times.

**Figure 3 ijms-23-05225-f003:**
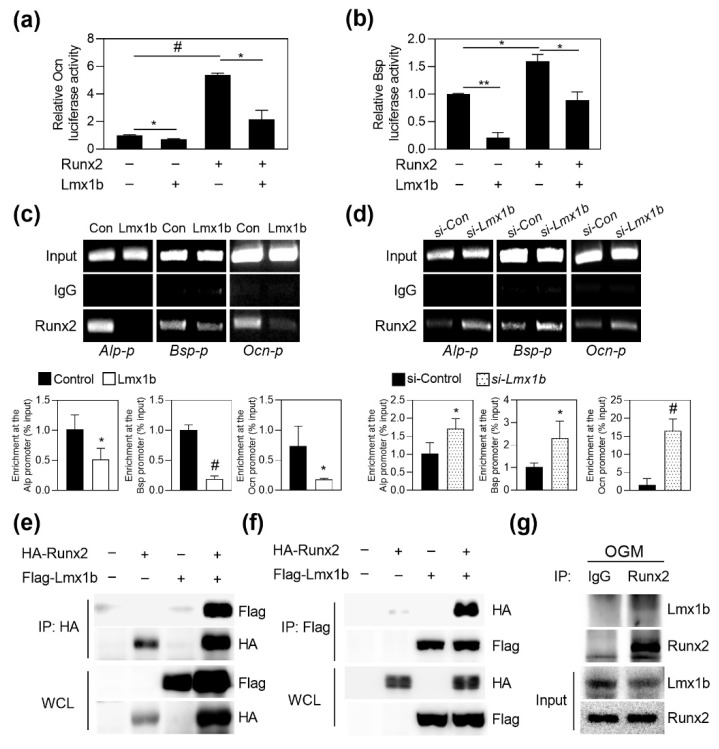
Lmx1b physically interacts with Runx2 to suppress its transcriptional activity. (**a**,**b**) 293T cells were transfected with the Ocn or Bsp luciferase reporter in the presence or absence of Runx2 and/or Lmx1b. After 48 h of transfection, the cells were assayed for relative luciferase activity. Data represent the means ± SD of triplicate samples. * *p* < 0.05, ** *p* < 0.001, # *p* < 0.005 vs. mock. Results are representative of experiments that were independently repeated at least three times. (**c**) Primary osteoblast precursor cells transduced with either pMX-IRES-EGFP (control) or Lmx1b retroviruses were cultured in OGM for four days and subjected to chromatin immunoprecipitation. Immunoprecipitation was performed with control IgG or anti-Runx2. Precipitated DNA was subjected to ChIP-qPCR with primers specific to the Runx2-binding site within Alp, Bsp, or Ocn promoter. Data represent the means ± SD of triplicate samples. * *p* < 0.05 and # *p* < 0.005, vs. Control. (**d**) Primary osteoblast precursor cells transfected with control siRNA (si-Control) or siRNA specific for Lmx1b (*si-Lmx1b*) were cultured in OGM for four days and subjected to chromatin immunoprecipitation. Immunoprecipitation was performed with control IgG or anti-Runx2. ChIP-qPCR was performed using precipitated DNA with primers specific to Runx2-binding sites within Alp, Bsp, or Ocn promoter. Data represent the means ± SD of triplicate samples. * *p* < 0.05 and # *p* < 0.005, vs. si-Control. (**e**–**g**) 293T cells were cotransfected with Flag-Lmx1b and HA-Runx2. Cell lysates were subjected to reciprocal co-immunoprecipitation with anti-HA (**e**) or anti-Flag (**f**). Interaction between Runx2 and Lmx1b was analyzed by western blotting using an anti-HA or anti-Flag. (**g**) Primary osteoblast precursor cells were cultured for four days in OGM. Cell lysates were immunoprecipitated with an anti-Runx2. The interaction between endogenous Runx2 and Lmx1b was analyzed by western blotting using an anti-Lmx1b or anti-Runx2.

**Figure 4 ijms-23-05225-f004:**
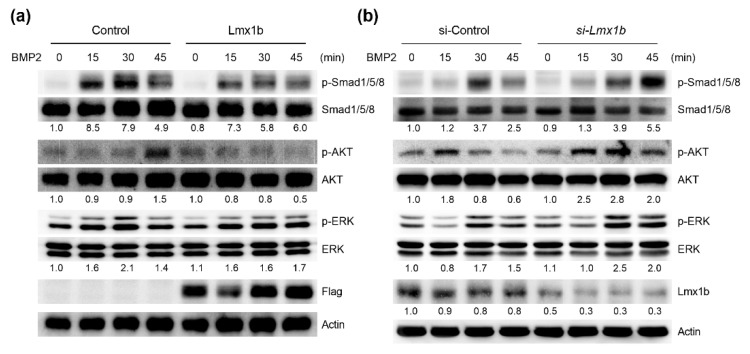
Lmx1b regulates BMP2 signaling pathways in osteoblasts. (**a**) Primary osteoblast precursor cells transduced with pMX-IRES-EGFP (Control) or Lmx1 retroviruses were stimulated with bone morphogenic protein 2 (BMP2) for the indicated times. Whole cell lysates were subjected to western blot analysis with specific antibodies, as indicated. The band intensity of p-Smad1/5/8, p-AKT and p-ERK were normalized to correspond with Smad1/5/8, AKT, and ERK, respectively, and number represents the densitometry ratios. (**b**) Primary osteoblast precursor cells transfected with control siRNA (si-Control) or siRNA specific for Lmx1b (*si-Lmx1b*) were stimulated with BMP2 for the indicated times. Whole cell lysates were subjected to western blot analysis with specific antibodies, as indicated. The band intensity of p-Smad1/5/8, p-AKT, p-ERK, and Lmx1b were normalized to correspond to Smad1/5/8, AKT, ERK, and Actin, respectively, and number represents the densitometry ratios.

**Figure 5 ijms-23-05225-f005:**
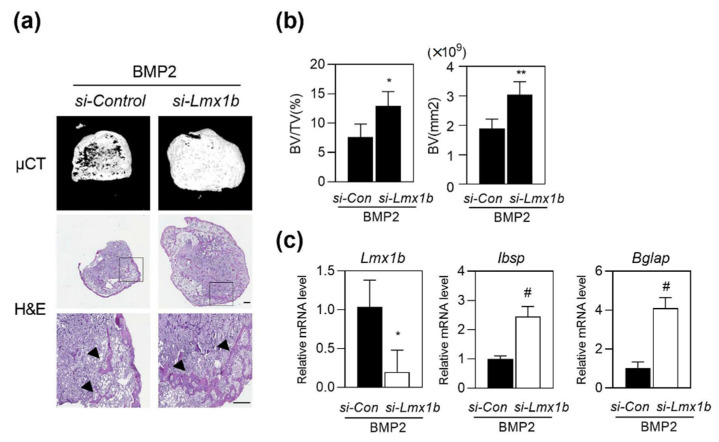
Silencing of *Lmx1b* promotes ectopic bone formation in vivo. Collagen sponges absorbed with control siRNA (si-Control) or siRNA specific for Lmx1b (*si-Lmx1b*) and BMP2 (1 µg) were subcutaneously implanted (*n* = 7 per group). (**a**,**b**) Ectopic bones were biopsied and subjected to micro computed tomography (µCT) and histology. (**a**) Representative 3D images of ectopic bone formation analyzed by µCT (upper panel). Hematoxylin and eosin staining of biopsied ectopic bones (lower panel). The arrow head indicates newly formed bones. Scale bars = 300 μm. (**b**) Bone volume per tissue volume (BV/TV) and bone volume (BV) of biopsied ectopic bones were assessed using µCT. * *p* < 0.05, ** *p* < 0.001 vs. si-Control with BMP2. (**c**) Total RNA was isolated from ectopic bone specimens, and mRNA expression of Lmx1b, Ibsp, and Bglap was assessed using quantitative real-time PCR. Data represent the means ± SD of triplicate samples. * *p* < 0.05, # *p* < 0.005 vs. si-Control with BMP2. Results are representative of experiments that were independently repeated at least three times.

## Data Availability

The datasets generated for this study are available on request to the corresponding author.

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
