# Peer review of "Transcription Factor Lmx1b Negatively Regulates Osteoblast Differentiation and Bone Formation"

_ijms, 2022, doi:10.3390/ijms23095225_

Round 1

Reviewer 1 Report

It does not get really clear how LmxB1 impacts the phosphorylation of downstream BMP signaling mediators like Smads etc. Does LmxB1 interact with the BMP receptors or how do those affected phosphorylation levels get achieved in the experiments? Could the authors please comment on this a bit in more detail within the discussion section?!

Also, could the authors please explain a bit clearer why they think that physical LmxB1-Runx2 interaction prevents Runx2 from proper binding to OSE2 elements. I think a reasonable alternative scenario would be that LmxB1 simply exhibits repressor activities when bound to Runx2 while this is attached to a dedicated OSE2 site.

Reviewer 2 Report

I am not able to find any description of the methods used to perform osteoclastogenesis as well as the figures relative to the results described in the paragraph 2.1

The figure 2b is not clear since there is a portion of the well not covered by cells (Si-control OGM+). The quantification does not reflect the alizarin red staining.

All the experiments using murine cells. It should be better to use human osteoclast precursors to confirm the results also in a human model.

The results reported in the Figure 4 are not clear. Is the control w/o BMP=1? Is the fold change between control and Lmx1b different with and without BMP2?

Round 2

Reviewer 2 Report

In the paragraph 2.4, the authors claim that the Smad1/5/8, AKT, and ERK signaling pathways were inhibited in osteoblasts infected with Lmx1b overexpression. Looking at the fold change of Fig 4, this is true for Smad 15-30', akt 45', but it is not so obvious for erk. the authors should at least discussion the significance of this different response. Moreover, the word "inhbition" is not appropriate since the fold change for smad and erk is still very high. I think the words "blunted, attenuated" are more appropriate. 

Yet, in the figure 4b the inhibition of Lmx1b is not evident (w/o BMP2 the fold change of Lmx1b is 0.8 compared to the si-Cnt). Hoe did they check the inhibition? In the same way the stimulatory effect with and w/o BMP2 is not clear (especially for erk)

Author Response

In the paragraph 2.4, the authors claim that the Smad1/5/8, AKT, and ERK signaling pathways were inhibited in osteoblasts infected with Lmx1b overexpression. Looking at the fold change of Fig 4, this is true for Smad 15-30', akt 45', but it is not so obvious for erk. the authors should at least discussion the significance of this different response. Moreover, the word "inhbition" is not appropriate since the fold change for smad and erk is still very high. I think the words "blunted, attenuated" are more appropriate. 

Response: Although BMP2 activated several signaling pathways, in the present study ERK activation was not effectively induced by BMP2 compared to that of other signaling pathways. Therefore, magnitude of inhibition or elevation of ERK activation by Lmx1b overexpression or Lmx1b silencing, respectively, was not immense as those of other signaling pathway. We added discussion part.

We replaced “inhibited” with “attenuated” in the results as suggested.

Yet, in the figure 4b the inhibition of Lmx1b is not evident (w/o BMP2 the fold change of Lmx1b is 0.8 compared to the si-Cnt). Hoe did they check the inhibition? In the same way the stimulatory effect with and w/o BMP2 is not clear (especially for erk)

Response: Fold changes were calculated by densitometry. Intensities of Lmx1b and other phosphorylated forms of signals were normalized by actin and their total forms, respectively. The calculated values are depicted below the bands in the figure 4. As you pointed the inefficiency of silencing Lmx1b and ERK activation, we repeated experiments to confirm the signaling pathways, and we observed that Lmx1b was downregulated from 30% to 50%, and we replaced the Figure 4b.

Round 3

Reviewer 2 Report

Thank you of your response